# A Multimodal Bracelet to Acquire Muscular Activity and Gyroscopic Data to Study Sensor Fusion for Intent Detection

**DOI:** 10.3390/s24196214

**Published:** 2024-09-25

**Authors:** Daniel Andreas, Zhongshi Hou, Mohamad Obada Tabak, Anany Dwivedi, Philipp Beckerle

**Affiliations:** 1Chair of Autonomous Systems and Mechatronics, Friedrich-Alexander-Universität Erlangen-Nürnberg, 91054 Erlangen, Germanyphilipp.beckerle@fau.de (P.B.); 2Artificial Intelligence (AI) Institute, Division of Health, Engineering, Computing and Science, University of Waikato, Hamilton 3216, New Zealand; 3Department of Artificial Intelligence in Biomedical Engineering, Friedrich-Alexander-Universität Erlangen-Nürnberg, 91054 Erlangen, Germany

**Keywords:** wearable device, electromyography, force myography, sensor fusion, muscular activity, robotic hand control

## Abstract

Researchers have attempted to control robotic hands and prostheses through biosignals but could not match the human hand. Surface electromyography records electrical muscle activity using non-invasive electrodes and has been the primary method in most studies. While surface electromyography-based hand motion decoding shows promise, it has not yet met the requirements for reliable use. Combining different sensing modalities has been shown to improve hand gesture classification accuracy. This work introduces a multimodal bracelet that integrates a 24-channel force myography system with six commercial surface electromyography sensors, each containing a six-axis inertial measurement unit. The device’s functionality was tested by acquiring muscular activity with the proposed device from five participants performing five different gestures in a random order. A random forest model was then used to classify the performed gestures from the acquired signal. The results confirmed the device’s functionality, making it suitable to study sensor fusion for intent detection in future studies. The results showed that combining all modalities yielded the highest classification accuracies across all participants, reaching 92.3±2.6% on average, effectively reducing misclassifications by 37% and 22% compared to using surface electromyography and force myography individually as input signals, respectively. This demonstrates the potential benefits of sensor fusion for more robust and accurate hand gesture classification and paves the way for advanced control of robotic and prosthetic hands.

## 1. Introduction

People with upper-limb amputation rely on robust control of the prosthesis [1]. Even with modern techniques, users can still not control the prosthesis as well as their own limbs [1]. In the past, researchers tried to restore hand functionality by using biosignals to control robotic and prosthetic hands. Surface electromyography (sEMG), which records electrical muscle activity using non-invasive electrodes [2], has been the primary method used in most studies [3,4,5,6]. sEMG is also frequently used to teleoperate robotic hands in both virtual and real-world settings [3,7,8,9]. However, the limitations of sEMG-based control systems have contributed to low acceptance of myoelectric prostheses among people with amputation [10,11]. Additionally, the control of robotic and virtual hands is often hindered by the limited capabilities and low robustness of single input signals. Therefore, to achieve dexterous control of devices, it is essential to develop frameworks that combine sEMG-based control with other sensing modalities, creating more sophisticated and robust control algorithms [12,13].

sEMG-based methods are non-stationary but prone to crosstalk between different muscles and sensitive to electrode shifts during their use, sweating, fatigue, and electromagnetic noise [14,15]. While there are alternatives to sEMG to be used for hand gesture classification, they come with their own downfalls. Mechanomyography (MMG), for instance, which measures muscle oscillations through accelerometers or microphones [16], is robust to interferences of the skin–sensor interface but at the same time is prone to crosstalk between different muscle groups, similar to sEMG. Furthermore, MMG is susceptible to interference from ambient acoustic and vibrational noise [17,18]. Another common method to acquire muscular activity is through Force Myography (FMG), also referred to as muscle pressure mapping [19], measuring volumetric changes of the limb due to muscle contractions [20]. Compared to sEMG, FMG is easy to use, provides a stable signal over time, and is cost-efficient [20,21] but lacks high-frequency information and is prone to disturbances through external forces. Further, FMG has successfully been used as an input signal for the classification of hand gestures in previous works [20,21,22,23,24].

Previous works have already shown the benefits of combining sensing modalities to improve the classification accuracy of hand gestures from muscular activity [25]. A common fusion of modalities is between sEMG and Inertial Measurement Unit (IMU) data [25]. The gyroscopic data from the IMU could compensate for electrode shifts during arm movements, while the accelerometer can acquire Mechanomyography (MMG) data at sufficient sampling rates of around 200 Hz [25]. The combination of sEMG and FMG has also been investigated in the past; an overview can be seen in Table 1. Previous studies that compared the classification accuracy using sEMG and FMG as input data have seen improvements in using the combined signal over each individual signal [14,15,26,27,28,29]. Jiang et al. [14], for instance, noticed an improvement in the classification accuracy of hand gestures by 10% using both input signals combined compared to using each signal individually.

There are different ways to locate sEMG and FMG sensors on the forearm. While in some studies sEMG and FMG sensors were placed in different locations on the forearm (off-located sensor configuration) [15,29,30,31], others combined both modalities into a single module (co-located sensor configuration) [12,14,26,27,28]. The latter approach is advantageous because it allows for direct comparison of the signals and conserves space, which is crucial for people with amputation who have a short residual limb. As shown in Table 1, the most common sensors used in combination with sEMG are Force-Sensitive Resistors (FSRs) [12,15,26,28,29,30,31], which change their resistance based on the applied force. However, there is no consensus as to which method is most suitable to combine sEMG with FMG.

As Table 1 shows, there was previously only one device that could simultaneously capture sEMG, FMG, and IMU data [12]. However, the device by Gharibo et al. [12] only contains a single IMU instead of one for each module. Integrating all sensing modalities into each module would allow for a direct comparison between them and demonstrate the potential benefits of their combined use for detecting user intent. A multimodal bracelet that captures muscular activity would also provide sensor redundancy, enabling more reliable control of robotic hands. Most existing devices that combine sEMG and FMG have custom-designed sEMG sensors [12,14,26,27,28,29], as shown in Table 1. However, the commercial Trigno Avanti sEMG sensors by Delsys Incorporated (Natick, MA, USA), which also contain a six-axis IMU, are known for their high quality and have been widely used in various studies [11,32,33,34] and were the primary choice in the Ninapro database [35].

Therefore, instead of designing new sEMG electrodes, we aim to leverage the Trigno Avanti sensors. Our goal is to develop an FMG system that can be combined with these Trigno Avanti sensors into a single module, enabling accurate mapping of muscle contractions. Furthermore, we aim for a compact and wireless design, making the device easy to use. Additionally, the bracelet shall fit a variety of different arm circumferences to allow for compatibility across a wide population.

## 2. Design of the Multimodal Bracelet

The multimodal bracelet contains five identical sub-modules and one main module, as illustrated in Figure 1. The modules are connected by flexible links made of Thermoplastic Polyurethane (TPU) to provide modularity and flexibility (see Figure 1 on the right), making the bracelet suitable to be worn on both the forearm and upper arm.

The design of a sub-module is illustrated in Figure 2. Each sub-module is housed in a 3D-printed case made of Polylactic Acid (PLA). Inside, there is a Printed Circuit Board (PCB) with four Force-Sensitive Resistors (FSRs), one at each corner, and a Trigno Avanti sEMG sensor by Delsys Incorporated, with a three-axis accelerometer and a three-axis gyroscope. The PCB with the FSRs fits into a groove inside the sub-module cover. The electrodes at the bottom of the sEMG sensor must contact the skin to ensure good signal quality. Therefore, these electrodes extend 3 mm outside the case when worn, as shown in the right picture of Figure 1. Protrusions on the case edge limit the downward movement of the sEMG modules to prevent them from falling out, but they allow vertical movements to transmit force to the FSRs above, as depicted in Figure 2. When the armband is worn, muscle contractions cause the skin to press the sEMG sensor against the interior of the case. Four rubber feet are positioned at the corners of a 3D-printed structure that acts as a force transmitter located on top of the sEMG module, aligning precisely with the FSR sensors on the PCB (see Figure 2). This setup ensures reliable force transmission from the sEMG sensor to the FSRs and was inspired by the work of Ke et al. [28]. However, instead of using a single FSR per module, each module now contains four FSRs (one in each corner), providing a more accurate mapping of the muscle motions inside the user’s forearm. With six modules in total, this design includes six sEMG channels, each with a six-axis IMU, and 24 FMG channels, resulting in comprehensive and precise data collection.

The sub-modules transmit the signal received by the FSRs to the main PCB located on the main module. The main module is divided into two parts: the lower half is identical to the sub-modules as shown in the upper picture of Figure 3, while the upper half is a flat rectangular box that accommodates the main PCB and provides space to house a battery underneath to allow complete wireless operation, which is shown in the lower pictures of Figure 3.

The PCB layout of the bracelet is shown in Figure 4. The design is based on the ESP32-S2-Saola-1 board with the ESP32-S2-WROVER wireless module by Espressif Systems, Shanghai, China. It contains an Xtensa single-core 32-bit LX7 microprocessor that clocks up to 240 MHz and an onboard PCB antenna that can transfer data at a bit rate of up to 150 Mbps. The board can be flashed and powered through a micro USB port. To allow fully wireless operation, the microcontroller can alternatively be powered by a 3.7 V lithium polymer battery through the respective connectors shown in Figure 4. The FSRs of the type 400 Short (Interlink Electronics Inc., Camarillo, CA, USA) can sense forces in the range of 0.1 N to 10 N and were connected to the analog inputs of the ESP32-S2-Saola-1 board and the CD74HC4067 Multiplexer (Texas Instruments Incorporated, Dallas, TX, USA). The FSR signals can be acquired at a rate of 20 Hz through a voltage divider and stabilized by an MCP6044 operational amplifier (Microchip Technology Inc., Chandler, AZ, USA). The sensor reading measured in Volts is an arbitrary value that can be calibrated to measure force in Newtons if needed. The design of the multimodal bracelet can be customized and easily manufactured using 3D printing to allow compatibility with other commercial and non-commercial sEMG sensors.

## 3. Functional Tests and Evaluation

The primary objective of this experiment was to validate the signal quality of the bracelet by classifying hand gestures from the acquired data using a machine learning algorithm. The focus was mainly on the self-built FMG, since the sEMG and IMU data are acquired by the commercial Trigno Avanti sEMG sensors by Delsys Incorporated. Furthermore, we aimed to explore the potential benefits of fusing sEMG, FMG, and IMU data for gesture classification. Data were acquired from five male participants, of which two are authors of this paper. The average age of the participants was 27.20±2.77. The participants were asked to complete 20 rounds of gesture sequences while wearing the bracelet on their forearm as in Figure 5.

Each round consisted of five different gestures: rest, pinch, tripod, fist, and stretch, as shown in Figure 6. The order of these gestures was randomized, and participants followed computer prompts to perform the corresponding actions. The participants performed the gestures in direct sequence without resetting to a resting position. Each gesture was performed within a time window of five seconds. The first two seconds of each gesture were counted towards the participant’s reaction time and the transition and completion of the current gesture. The remaining three seconds were acquired for the dataset and automatically labeled according to the computer prompts.

The raw signals from the six sEMG sensors were collected at a sampling frequency of 2000 Hz. IMU and FSR data were collected at sampling rates of 200 Hz and 10 Hz, respectively, and then upsampled to match the 2000 Hz frequency of the sEMG sensors. Signals from all sensors were acquired as 32-bit floating-point values. The measurements by the sEMG and FSR sensors were acquired in Volts, while the quaternions by the IMU are provided as gravitational force equivalents. In this experiment, the IMU acquired quaternions only, providing an exact representation of the orientation. Each data point was labeled with the corresponding gesture. The complete dataset was divided into windows, with each window containing 500 data points (250 ms) and overlapping by 473 data points (236.5 ms). The features extracted from each window for each signal were adapted from [14,26,28]:sEMG: Mean Absolute Value (MAV), Mean Absolute Value Slope (MAVS), Zero Crossings (ZCs), Slope Sign Changes (SSCs), Wavelength (WL), and Root Mean Square (RMS);FMG: Mean Value (MV) and RMS;IMU: MV.

The resulting length of the feature vector (number of features × number of channels) was 36 for sEMG, 48 for FMG, and 24 for IMU data. The multiple signal modalities were then fused by concatenating all computed features of each input signal into a single feature vector. The resulting feature vector was then used as input to train a random forest (RF) classifier, which combines the outputs of multiple decision trees trained on different subsets of data and features [36,37] to decode muscular activity into hand gestures. Other established classification models, such as Support Vector Machines (SVMs), K-Nearest Neighbors (KNN), or neural networks, were considered. However, random forests were preferred for their efficiency in training and prediction, robustness to signal noise, and ability to handle large feature spaces without extensive parameter tuning [38,39]. Overall, random forests provide a balanced trade-off between accuracy and computational efficiency, making them a suitable choice for this analysis, and have already demonstrated their effectiveness for the classification of hand gestures from muscular activity in previous studies [40,41]. The parameters used to train the RF model include 50 estimators, a maximum tree depth of 50, a minimum of 5 samples per leaf node, and a minimum of 15 samples required to split a node. The acquired data were evaluated using K-fold cross-validation with 20 folds for each participant. Thus, the RF model was trained on 19 rounds of gesture sequences and tested on the remaining one. This process was repeated until each gesture sequence was used once as a testing set and the overall accuracy was averaged across the 20 folds. This process was repeated separately for the inputs from sEMG, FMG, and IMU and for each combination of the input signals. The results were averaged across all five participants and are displayed as confusion matrices in Figure 7 and Figure 8.

Using FMG as the input signal resulted in a higher average classification accuracy across participants of 90.1±4.5% compared to EMG reaching 87.8±6.3%. Among the three input signals, the orientation data of the IMU yielded the worst results with an accuracy of 64.0±5.3%.

Figure 8 shows that the combination of sEMG + IMU and FMG + IMU led to lower classification accuracies of 85.0±5.1% and 89.5±3.3%, respectively, compared to using sEMG and FMG data individually as the input signal. The combination of sEMG + FMG, however, led to a higher accuracy of 91.9±3.0% compared to each of the individual input signals. This was only superpassed by combining all three input signals sEMG + FMG + IMU, which yielded a classification accuracy of 92.3±2.6%.

Figure 9 shows the acquired signals from the second trial of participant 1 over time. The upper plot shows the muscle activation measured by the sEMG electrodes, the middle plot the quaternions measured by one of the six modules, and the lower plot one FSR value from each module. The gray shaded areas mark the 2 s transition times between gestures that were disregarded for model training and testing. The signals show distinct changes across different gestures, especially for sEMG and IMU data.

## 4. Discussion

The confusion matrices in Figure 7 and Figure 8 reveal that the tripod and pinch gestures have the lowest classification accuracies for all input signal combinations that include sEMG. These two gestures are rather often confused with each other, which is likely caused by the similarities between the two motions and overlapping muscles that lead to crosstalk between different muscle groups, thus resulting in similar sEMG signals, which can also be seen in Figure 9. Similar behavior has been reported by Kaczmarek et al. [42], where the pinch gesture with the index finger and the pinch gesture with the middle finger were classified with the lowest accuracies across all gestures. Using FMG as an input signal for the RF model, on the other hand, shows reduced classification accuracies for the pinch and rest gesture. This highlights the difference in information between sEMG and FMG and hints at the potential benefit of fusing both signals. The results from Figure 7 and Figure 8 confirm this assumption, showing that the fusion of sEMG and FMG data as input signals for an RF model is especially beneficial for classifying different hand gestures, which is in line with previous studies [14,15,26,27,28,29]. Interestingly, the combinations of sEMG + IMU and FMG + IMU yield lower classification accuracies than sEMG and FMG individually. The IMU data likely contain noise or at least partially irrelevant data for this task. Looking at Figure 9, there appears to be less information in the IMU data overall compared to the other two modalities, which is mainly due to the absence of arm movements during the task execution. The added noise or irrelevant data of the IMU signal seem to outweigh any additional beneficial information, thus causing a lower classification accuracy in combination with sEMG or FMG. This effect might be reduced by increasing the size of the RF model, making it less prone to noise or irrelevant data through the larger number of decision trees. Furthermore, gyroscopic data of the IMU are expected to be more beneficial when arm motions are included, which can lead to electrode shift and thus misclassifications when using sEMG and FMG data only. Gharibo et al. [12] already tested combinations with IMU data for different arm orientations and achieved good results, but they did not directly compare the accuracy to the individual signals, which asks for further investigation.

The maximum classification accuracy of 92.3±2.6% across all participants was achieved by using sensor fusion of all input signals. The accuracy appears to be low compared to previous studies [14,15,26,27,28,29,43]. Ovadia et al. [43], for instance, reached a classification accuracy of 99% across five subjects classifying six hand gestures, each repeated 30 times. They achieved this using a more sophisticated machine learning algorithm compared to the RF model used in this study. The low classification accuracy can be explained by multiple factors. One of them is how closely the participants followed the instructions during the experiment. Any falsely performed gesture would cause mislabeling in the dataset. By analyzing the results of each participant more closely, we found that in some of the folds of the K-fold cross-validation, the classification accuracy of individual gestures was close to 0%. These gestures were often confused almost 100% with another gesture. This is unlikely caused by a low performance of the RF model but rather due to participants performing a different gesture to the one being prompted on the screen. This could be prevented by closely monitoring the actions performed by the participants and repeating any falsely performed gesture. Moreover, the participants performed the gestures in direct sequence without resetting to a resting position. While this can be closer to real-life scenarios, it requires higher concentration by the participants and likely causes muscle fatigue throughout the experiment, which negatively impacts the signal quality.

The goal of this experiment, however, was not to reach maximum decoding accuracies, but to test the functionality of the proposed bracelet that combines multiple modalities, co-located in each module, to explore the potential of sensor fusion for the classification of hand gestures. The results clearly show the functionality of the bracelet, with the self-built 24-channel FMG system yielding better results than the commercial sEMG sensors regarding the classification accuracy of hand gestures using the same surface area on the forearm for data acquisition. This might be due to the larger number of FMG channels, potentially providing a better representation of muscle activation. Combining all three input signals could effectively reduce misclassifications by 37% and 22% compared to using sEMG and FMG individually as input signals for the RF classifier, respectively, highlighting the potential of sensor fusion for hand gesture classification.

## 5. Conclusions

In this work, we proposed and investigated a multimodal bracelet that combines six commercial sEMG sensors that each contain a six-axis IMU with a 24-channel FMG system into a single, wearable, and wireless device to acquire muscular activity for intent detection. The functional tests, which involved decoding hand gestures from five participants using a random forest model, yielded promising results. The data of the self-built FMG system led to a higher classification accuracy compared to the commercial sEMG sensors using the same surface area on the forearm for data acquisition. Overall, the results indicate an improvement in classification accuracy through the sensor fusion of multiple input signals, which is in line with previous studies [12,14,15,26,27,28,29,30,31]. For a deeper analysis of the benefits of sensor fusion for hand gesture classification, the experiment of this work should be repeated with more participants and a variety of sufficiently large machine learning models that are optimized for each combination of input data. Future experiments could also investigate the added value of accelerometer data for sensor fusion, which was not acquired in this initial evaluation, to improve the classification of hand gestures. These experiments can then be extended towards regression to continuously predict finger joint angles to allow for more complex and realistic hand motions.

We have made the plans for all mechanical and electronic components publicly available on GitHub (https://github.com/ASM-FAU/Multimodal-Bracelet, accessed on 30 June 2024). This enables researchers to rebuild the device or adjust the bracelet to fit other sEMG sensors, facilitating further exploration of the benefits of sensor fusion for hand gesture classification. Future research could focus on testing the robustness of classification in online real-time experiments using multiple input signals, which could also provide a redundancy in case a sensor fails. The proposed bracelet can be used to collect extensive datasets of daily life activities. This could help identify which specific motions benefit most from fusing specific input signals, and aid in finding the ideal sensor combination for robust intent detection to control robotic and prosthetic hands in the virtual and real world. Such findings could enable decoding complex finger motions rather than predefined gestures, advancing the control of robotic hands to be more similar to human hand movements.

## Figures and Tables

**Figure 1 sensors-24-06214-f001:**
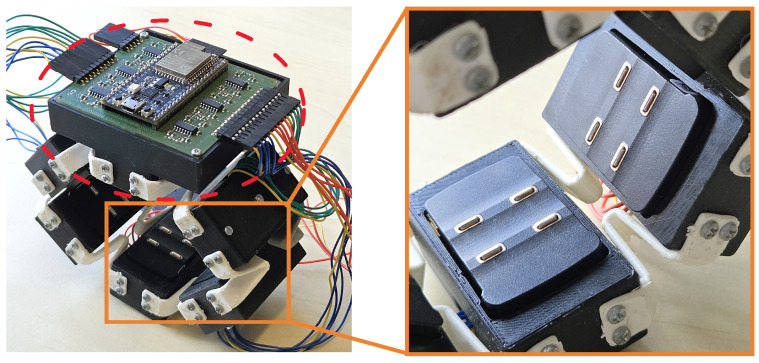
Pictures of the bracelet showing the overall design with the main module marked by the red dashed circle (**left**) and a detailed view of the sub-modules that are connected by white flexible links (**right**). The sEMG sensors protrude the cases by 3 mm to ensure good contact of the electrodes with the user’s skin.

**Figure 2 sensors-24-06214-f002:**
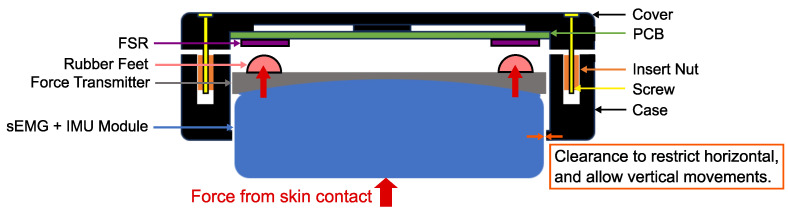
Schematic of a sub-module, housing a Trigno Avanti sEMG sensor from Delsys, Natick, USA, with an integrated 6-axis IMU, which can move vertically to transmit force to the four FSRs above.

**Figure 3 sensors-24-06214-f003:**
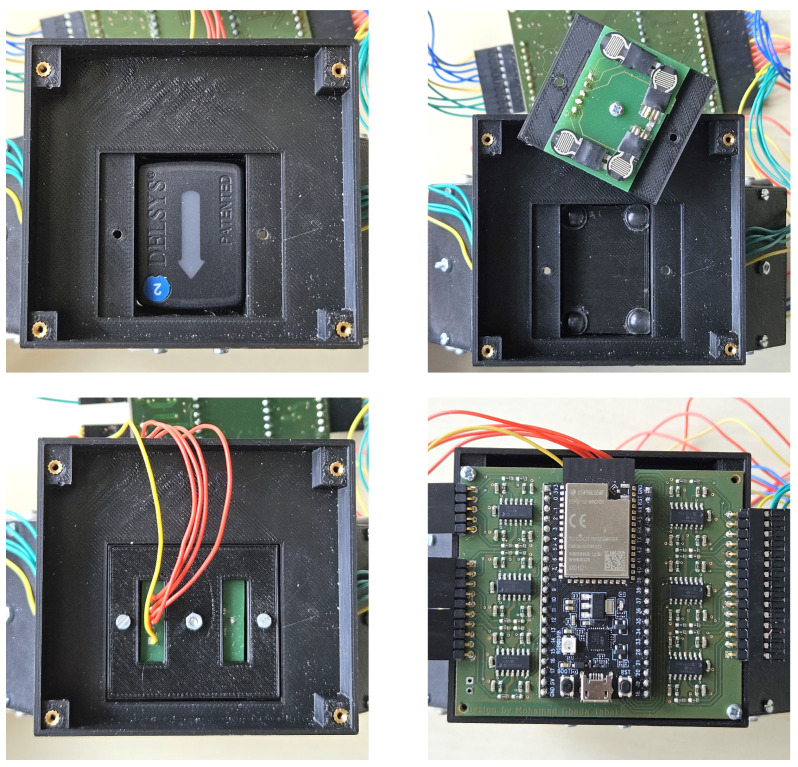
Pictures showing the structure of the main module of the bracelet. The lower half is identical to the sub-modules and contains an sEMG sensor and the PCB with FSRs in each corner, as shown in the upper pictures. The upper half of the main module provides space for a lithium polymer battery and the main PCB, as shown in the lower pictures.

**Figure 4 sensors-24-06214-f004:**
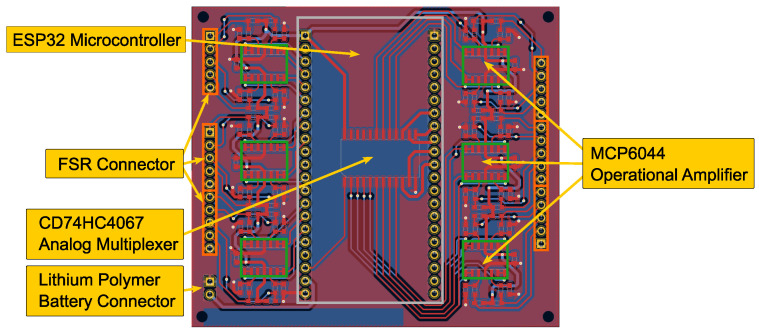
PCB layout with an ESP32 microcontroller and an analog multiplexer to allow data acquisition of 24 FSRs and wireless operation.

**Figure 5 sensors-24-06214-f005:**
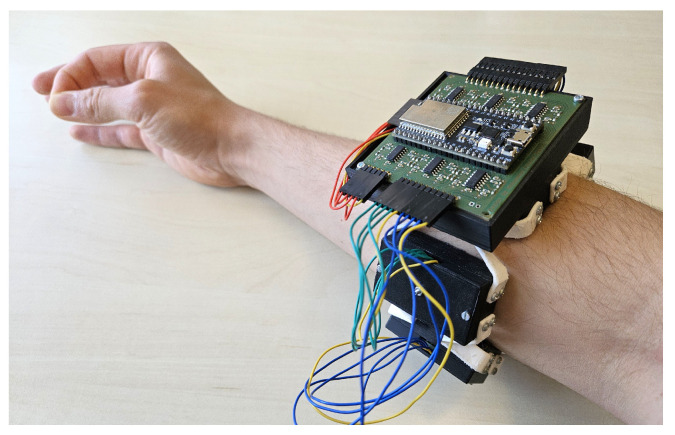
Experimental setup for data acquisition to test the functionality of the bracelet. The participants performed the gestures in a random order while wearing the bracelet on the forearm.

**Figure 6 sensors-24-06214-f006:**
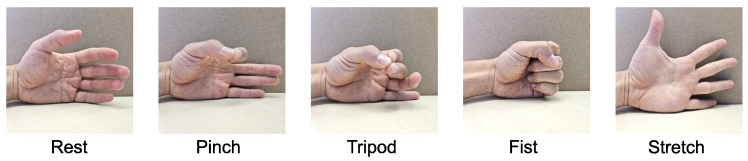
Gestures that were performed in a randomized order by the participants during data acquisition for 20 rounds.

**Figure 7 sensors-24-06214-f007:**
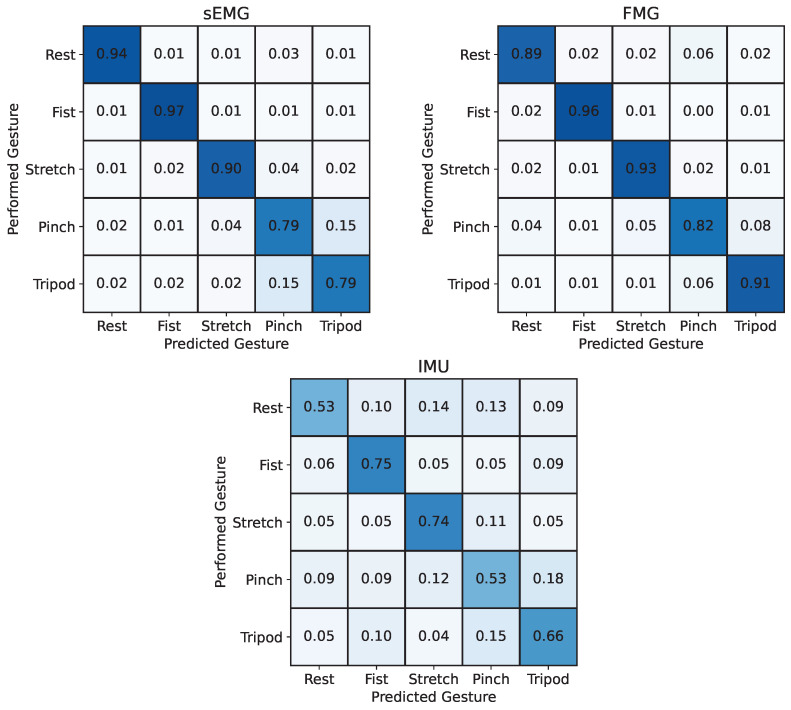
Confusion matrices from K-fold cross-validation averaged across all five participants for sEMG, FMG, and IMU as input signals decoded by a random forest classifier. Overall accuracies: sEMG: 87.8±6.3%, FMG: 90.1±4.5%, IMU: 64.0±5.3%. FMG yields better results than EMG and IMU as input signals for classifying five different hand gestures.

**Figure 8 sensors-24-06214-f008:**
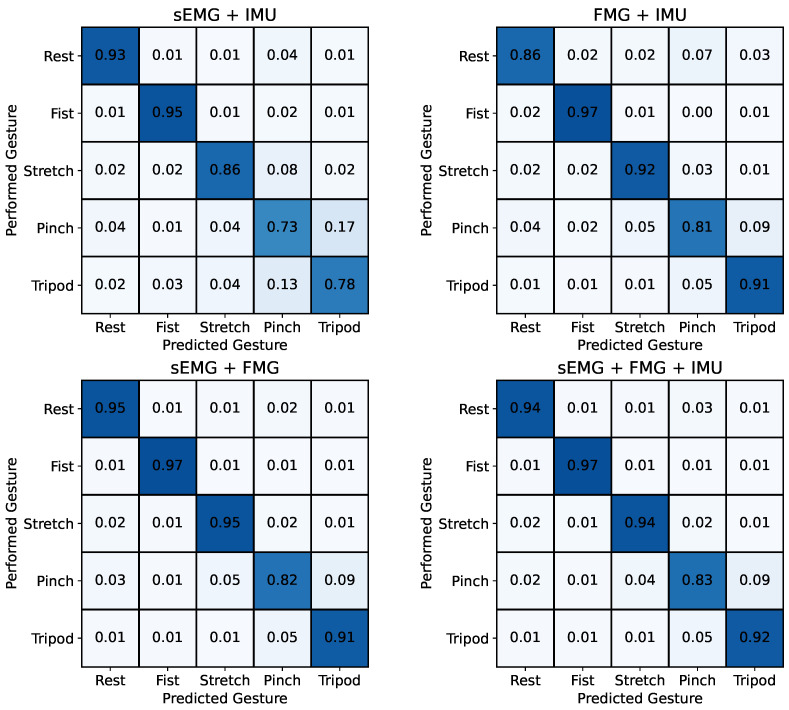
Confusion matrices averaged across all five participants for different combinations of input signals decoded by a random forest classifier. Overall accuracies: sEMG + IMU: 85.0±5.1%, FMG + IMU: 89.5±3.3%, sEMG + FMG: 91.9±3.0%, sEMG + FMG + IMU: 92.3±2.6%. The combinations of sEMG + IMU and FMG + IMU resulted in lower overall classification accuracies than using sEMG and FMG individually as input signals. The combination of all three input signals yielded the best results.

**Figure 9 sensors-24-06214-f009:**
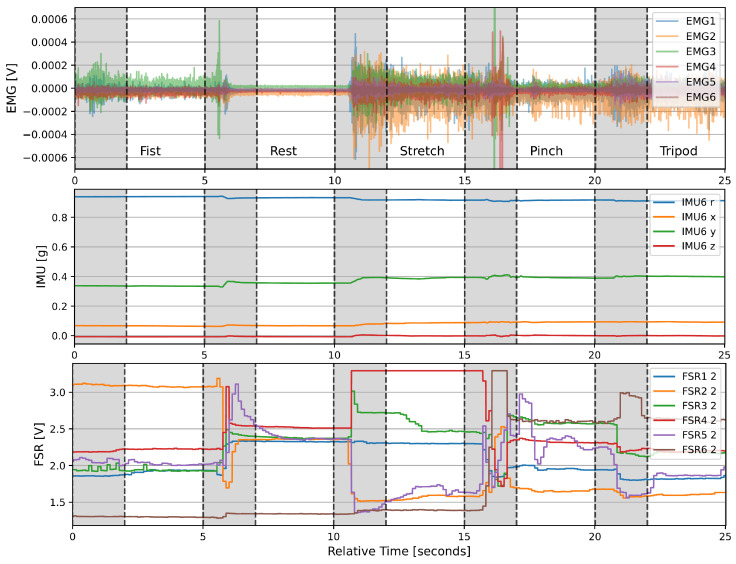
Time series plot of sEMG, IMU, and FSR signals acquired from participant 1 in the second trial. The gestures were indicated by the computer prompt in a random order. The transition times between gestures are shaded gray and were disregarded from training and testing the RF model. For better readability, IMU data were only plotted from one of the six modules, while FSR data were plotted for the second sensor of each module only.

**Table 1 sensors-24-06214-t001:** Studies combining sEMG, FMG, and IMU in one device, listing the types and number of sensors used. Parts of the table were adapted from the review paper of Zhou et al. [13].

Year and Reference	sEMG Sensor Type	No. of sEMG Channels	FMG Sensor Type	No. of FMG Channels	IMU Type	No. of IMU Channels	Co-Located Sensor Configuration *
2016 [15]	Ottobock MyoBock 13E200	10	FSR	10	none	0	no
2017 [30]	Ottobock MyoBock 13E200	2	FSR	90	none	0	no
2017 [31]	Ottobock MyoBock 13E200	10	FSR	10	none	0	no
2019 [29]	Wet silver electrodes	3	FSR	5	none	0	no
2020 [14]	Silver foil electrodes	8	Barometer	8	none	0	yes
2020 [28]	NeuroSky stainless steel electrodes	4	FSR	4	none	0	yes
2021 [26]	Silver-plated yarn	1	FSR	2	none	0	yes
2022 [12]	Convex electrodes	5	FSR	5	9-axis	1	yes
2023 [27]	Conductive silicon electrodes	3	Reflectance sensor	3	none	0	yes
This work	Delsys Avanti Trigno	6	FSR	24	6-axis	6	yes

* All sensors are considered co-located when they measure signals at the same muscle position [13].

## Data Availability

All the data are contained in the article.

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
