# Peer review of "A Multimodal Bracelet to Acquire Muscular Activity and Gyroscopic Data to Study Sensor Fusion for Intent Detection"

_sensors, 2024, doi:10.3390/s24196214_

Round 1

Reviewer 1 Report

Comments and Suggestions for Authors

In this work, the authors have proposed the design of a system for measuring multiple parameters related to myoelectric activity. The design consists of a structure that takes advantage of the shape and arrangement of the electrodes for EMG from a commercial system, adapting it to a myographic force-sensing system based on Force-Sensitive Resistors. Although the test approach is a well-studied topic today (hand gesture recognition), with impressive advancements based on different sensing and recognition techniques, it remains a current challenge to use myographic variables (electrophysiological and mechanical) in the optimization of these classifications. The hypothesis under which the authors propose this design is correct, as multiple variables measured in a system, frequently, and with the aid of current machine learning methods, lead to better classifications and clusterings of states.

The technical description of the design is adequate. The figures provided in this description are correct. I suggest the authors change the title of section 2 to 'Design of the Multimodal Bracelet.

It would be valuable for this work if the authors visualized the signals acquired in each sensing modality, not necessarily for all gestures, but for the most representative ones. In this context, I suggest the authors add a section following Section 2, for example, titled 'Multimodal Time Series.' This way, subsequent modifications that reference this work can compare the acquired signals. This would broaden the application field of the developed technology, such as in movement neuroscience and sports applications.

I suggest that the current section 3 'Functional Tests and Evaluation' be changed to section 4. Section 5 “Discussion” and Section 6 “Conclusion”

Minor:

Line 74. Change 'Table 5' to 'Table 1.

Author Response

In this work, the authors have proposed the design of a system for measuring multiple parameters related to myoelectric activity. The design consists of a structure that takes advantage of the shape and arrangement of the electrodes for EMG from a commercial system, adapting it to a myographic force-sensing system based on Force-Sensitive Resistors. Although the test approach is a well-studied topic today (hand gesture recognition), with impressive advancements based on different sensing and recognition techniques, it remains a current challenge to use myographic variables (electrophysiological and mechanical) in the optimization of these classifications. The hypothesis under which the authors propose this design is correct, as multiple variables measured in a system, frequently, and with the aid of current machine learning methods, lead to better classifications and clusterings of states.

Dear Reviewer, thank you for your comments. Below you find our replies.

Comment 1: The technical description of the design is adequate. The figures provided in this description are correct. I suggest the authors change the title of section 2 to 'Design of the Multimodal Bracelet.

Response 1: We have modified the title of section 2 accordingly.

Comment 2: It would be valuable for this work if the authors visualized the signals acquired in each sensing modality, not necessarily for all gestures, but for the most representative ones. In this context, I suggest the authors add a section following Section 2, for example, titled 'Multimodal Time Series.' This way, subsequent modifications that reference this work can compare the acquired signals. This would broaden the application field of the developed technology, such as in movement neuroscience and sports applications. I suggest that the current section 3 'Functional Tests and Evaluation' be changed to section 4. Section 5 “Discussion” and Section 6 “Conclusion”

Response 2: We added time series plots including all modalities from Trial 2 of Participant 1 to section 3 “Functional Tests and Evaluation” (page 10). The results are discussed in section 4 “Discussion”.

Comment 3: Line 74. Change 'Table 5' to 'Table 1.

Response 3: The Table number should now be correct.

Reviewer 2 Report

Comments and Suggestions for Authors

Pros

An innovative method to enhance the accuracy of automated hand motions in robots by analyzing various hand gestures captured from different sensors in an sEMG device. The authors have effectively designed and implemented the device, collecting multiple modalities of sensory data for further analysis.Points to consider

Points to consider

1.       When discussing different modalities, it would be helpful to explain the data formats collected from various sensors. If all signals from different sensors are converted to a uniform data type, what role does multimodality play?

2.       The authors could have elaborated on how they managed the different data formats, such as using a fusion algorithm, and how the data was decoded.

3.       Is there a specific reason for choosing the Random Forest classifier? Why were other popular classification models not considered?

4.       The authors could have strengthened their argument by comparing the classification results with other models to demonstrate the effectiveness of the Random Forest classifier.

5.       In the future, the device could be integrated into real-time robots to assess the actual performance of the proposed solution.

Comments on the Quality of English Language

It is reasonably good, but have a thorough review before publishing the corrected manuscript.

Author Response

An innovative method to enhance the accuracy of automated hand motions in robots by analyzing various hand gestures captured from different sensors in an sEMG device. The authors have effectively designed and implemented the device, collecting multiple modalities of sensory data for further analysis.

Dear Reviewer, thank you for your comments. Below you find our replies.

Points to consider 

Comment 1: When discussing different modalities, it would be helpful to explain the data formats collected from various sensors. If all signals from different sensors are converted to a uniform data type, what role does multimodality play?

Response 1: Since all signals are of the same data type, there was no need for a data type conversion. Thus, there should be no impact on multimodality.

We added the following to disclose the data type of the signals: “Signals from all sensors are acquired as 32-bit floating point values. The measurements by the sEMG and FSR sensors are acquired in Volts, while the quaternions by the IMU are provided as gravitational force equivalent.”

Comment 2: The authors could have elaborated on how they managed the different data formats, such as using a fusion algorithm, and how the data was decoded.

Response 2: There was no fusion algorithm used. The different signals were fused by concatenating the feature vectors of each signal. We changed the description to make this clearer:

“The multiple signal modalities are then fused by concatenating all computed features of each input signal into a single feature vector. The resulting feature vector was then used as input to train a Random Forest (RF) classifier, which combines the outputs of multiple decision trees trained on different subsets of data and features [37,38] to decode muscular activity into hand gestures.”

Comment 3: Is there a specific reason for choosing the Random Forest classifier? Why were other popular classification models not considered?

Response 3: We have considered other models but picked the Random Forest classifier for its efficiency and ease of use. This is what we added to the manuscript as explanation:

“Other established classification models, such as Support Vector Machines (SVM), K-Nearest Neighbors (KNN), or neural networks, were considered. However, Random Forests were preferred for their efficiency in training and prediction, robustness to signal noise, and ability to handle large feature spaces without extensive parameter tuning [39,40]. Overall, Random Forests provide a balanced trade-off between accuracy, and computational efficiency, making them a suitable choice for this analysis, and have already demonstrated their effectiveness for the classification of hand gestures from muscular activity in previous studies [41,42].”

Comment 4: The authors could have strengthened their argument by comparing the classification results with other models to demonstrate the effectiveness of the Random Forest classifier.

Response 4: The effectiveness of the Random Forest Classifier has already been demonstrated in previous studies [41,42]. The Random Forest Classifier is a common tool to classify muscular activity into hand gestures. Our work focuses on testing the functionality of the proposed device and the potential of sensor fusion for intent detection rather than comparing different classifiers to reach the highest possible classification accuracies.

Comment 5: In the future, the device could be integrated into real-time robots to assess the actual performance of the proposed solution.

Response 5: Thanks for the suggestion, we adjusted the Conclusion accordingly and framed it broader: “Future research could focus on testing the robustness of classification in online real-time experiments using multiple input signals, which could also provide redundancy in case a sensor fails.”

Round 2

Reviewer 1 Report

Comments and Suggestions for Authors

The authors have revised the manuscript in accordance with the suggestions I provided. In this context, I can recommend that this paper be published in this journal.